# Repeated Social Defeat Stress Induces HMGB1 Nuclear Export in Prefrontal Neurons, Leading to Social Avoidance in Mice

**DOI:** 10.3390/cells12131789

**Published:** 2023-07-05

**Authors:** Shiho Kitaoka, Ayaka Tomohiro, Shinya Ukeshima, Keyue Liu, Hidenori Wake, Shinya H. Kimura, Yasuhiko Yamamoto, Masahiro Nishibori, Tomoyuki Furuyashiki

**Affiliations:** 1Division of Pharmacology, Graduate School of Medicine, Kobe University, Kobe 650-0017, Japan; yakuri-at@med.kobe-u.ac.jp (A.T.); ecookjp@yahoo.co.jp (S.U.); 2Japan Agency for Medical Research and Development, Chiyoda-ku, Tokyo 100-0004, Japan; 3Department of Pharmacology, School of Medicine, Hyogo Medical University, Mukogawa-cho 1-1, Nishinomiya 663-8501, Japan; skimura@hyo-med.ac.jp; 4Department of Biological Science, Graduate School of Medicine, Kyoto University, Kyoto 606-8501, Japan; 5Department of Pharmacology, Graduate School of Medicine, Density and Pharmaceutical Sciences, Okayama University, Okayama 700-8558, Japan; liukeyue@md.okayama-u.ac.jp (K.L.); wake-h.kindai@med.kindai.ac.jp (H.W.); 6Department of Pharmacology, Faculty of Medicine, Kindai University, Sayama, Osaka 589-8511, Japan; 7Department of Biochemistry and Molecular Vascular Biology, Graduate School of Medical Sciences, Kanazawa University, Kanazawa 920-8640, Japan; yasuyama@med.kanazawa-u.ac.jp; 8Department of Translational Research and Drug Development, Graduate School of Medicine, Dentistry and Pharmaceutical Sciences, Okayama University, Okayama 700-8558, Japan; mbori@md.okayama-u.ac.jp

**Keywords:** repeated social defeat stress, depression, medial prefrontal cortex, HMGB1, RAGE

## Abstract

Inflammation has been associated with depression, and innate immune receptors, such as the Toll-like receptor (TLR) 2/4 in the medial prefrontal cortex (mPFC), are crucial for chronic stress-induced depression-related behaviors in mice. HMGB1, a putative ligand for TLR2/4, has been suggested to promote depression-related behaviors under acute stress. However, the roles of endogenous HMGB1 under chronic stress remain to be investigated. Here, we found that the cerebroventricular infusion of HMGB1 proteins blocked stress-induced social avoidance and that HMGB1-neutralizing antibodies augmented repeated social defeat stress-induced social avoidance in mice, suggesting the antidepressive-like effect of HMGB1 in the brain. By contrast, the infusion of HMGB1-neutralizing antibodies to the mPFC and HMGB1 knockout in α-CaMKII-positive forebrain neurons attenuated the social avoidance, suggesting the pro-depressive-like effect of HMGB1 released from prefrontal neurons under chronic stress. In addition, repeated social defeat stress induced HMGB1 nuclear export selectively in mPFC neurons, which was abolished in the mice lacking RAGE, one of HMGB1 receptors, suggesting the positive feedback loop of HMGB1-RAGE signaling under chronic stress. These findings pave the way for identifying multiple roles of HMGB1 in the brain for chronic stress and depression.

## 1. Introduction

Excessive or prolonged stress in adverse or demanding circumstances may induce depression and anxiety as well as cognitive impairments. Such maladaptive stress responses may precipitate or cause mental illnesses. However, how stress induces mental dysfunctions remains unclear. It has been reported that inflammatory cytokines are elevated in the blood from patients with depressive disorders [1,2]. Several groups including ours have identified the roles of inflammation-related molecules, such as IL-1β, TNF-α, and PGE_2_, for depression-related behaviors in a mouse model of chronic stress, which has often been used to study depression [3,4,5]. Studies using animal models of chronic inflammation have shown that pattern recognition receptors are crucial for inflammation without apparent infection. Pattern recognition receptors, including Toll-like receptors (TLRs), are thought to recognize endogenous ligands called damage-associated molecular patterns (DAMPs) released from cells upon damage to induce inflammation. Recently, we found that TLR2/4 in microglia in the medial prefrontal cortex (mPFC) is crucial for social avoidance, one of the depression-related behaviors, induced by repeated social defeat stress.

High-mobility group box 1 (HMGB1) is a DAMP that binds to TLR2/4. It was originally identified as a non-histone nuclear protein that interacts with and modulates the activities of transcription factors [6]. Tracey and his colleagues later showed that HMGB1 is released to the extracellular space from macrophages upon inflammatory stimuli, and that extracellular HMGB1 is crucial for endotoxin lethality in mice [7]. Previous studies using neutralizing antibodies to HMGB1 or its antagonists have indicated that extracellular HMGB1 is crucial for neuroinflammation and concomitant neuronal damages in animal models of neurological disorders and brain ischemia [8,9,10,11,12,13]. With these backgrounds, previous studies sought to examine the involvement of HMGB1 in depression-related behaviors. However, these studies have not proven the roles of endogenous HMGB1 in depression-related behaviors, as they only showed pro-depressive actions of exogenously applied HMGB1 and anti-depressive actions of non-selective inhibitors for HMGB1, such as glycyrrhizin, which binds to and inhibits not only HMGB1 but also other metabolic enzymes [14,15]. Furthermore, these studies only measured depression-related behaviors under acute stress and anhedonia even without stress. Thus, the roles of endogenous HMGB1 in chronic stress-induced depression-related behaviors remain unknown.

In the present study, we examined the roles of endogenous HMGB1 extracellularly released in the brain in repeated social defeat stress-induced social avoidance in mice.

## 2. Materials and Methods

### 2.1. Experimental Animals

We obtained 5-week-old male C57BL/6N mice and retired male ICR mice from Japan SLC. The mice were housed in an animal facility for at least one week after arrival. They were kept in a temperature- and humidity-controlled environment under 12 h light/12 h dark conditions with free access to a diet and water. αCaMKII-CreERT mice of a C57BL/6J background (B6;129S6-Tg(Camk2a-cre/ERT2)1Aibs/J) were purchased from Jackson Laboratories and maintained as heterozygous by backcrossing with C57BL/6N mice. HMGB1-flox (B6.129P2-Hmgb1tm1Ttg) mice were backcrossed with C57BL/6J mice [16]. Then, HMGB1-flox homozygous mice (HMGB1fl/fl) were obtained by heterozygous crossing and crossed with αCaMKII-CreERT heterozygous mice (αCaMKII-CreERT/+) to obtain HMGB1-flox homozygous littermates with (αCaMKII-CreERT/+; HMGB1fl/fl) or without (HMGB1fl/fl) the αCaMKII-CreERT allele. The mice were injected intraperitoneally with tamoxifen (T5648, Sigma Aldrich; 40 mg/kg, dissolved in 1:9 corn oil (C8267, Sigma Aldrich) and ethanol mixture) to obtain forebrain neuron-specific HMGB1 knockout or control mice, respectively. RAGE knockout (RAGE-KO) mice were backcrossed with C57BL/6J mice more than 10 times [17]. Then, RAGE-KO heterozygous mice were obtained by mating the homozygous mice and C57BL/6N mice, and the resultant heterozygous mice were crossed to obtain wild-type and homozygous littermates for experiments. All experiments were conducted in accordance with the institutional guidelines for the care and use of laboratory animals of Kobe University (protocol codes P150304 and P200201).

### 2.2. Repeated Social Defeat Stress and Social Interaction Test

We conducted repeated social defeat stress and the social interaction test as described previously [3,18]. For repeated social defeat stress, male ICR mice were chosen as aggressors based on their fighting intensity to male C57BL/6N mice. Male mice of 9–10 weeks old to be defeated (designated as the Defeat group) were isolated for one week and introduced in the home cage of a male ICR mouse used as an aggressor for 10 min daily for 10 consecutive days. A defeated mouse encountered a different aggressor mouse for each stress exposure to minimize the variability in the aggression level. Control mice (designated as the Naïve group) were placed in a novel cage without an ICR mouse instead.

For the social interaction test, one day before stress exposures (Day 0), the mice in the Naïve and Defeat groups were habituated for 150 s in an open field chamber (30 cm × 40 cm) with a mesh cage (10 cm × 6 cm) placed at one end. Then, before the first and seventh stress exposure (Day 1 and Day 7, respectively) and after the tenth, last stress exposure (Day 11), the mice were kept for 150 s in the open field chamber with an unfamiliar ICR mouse enclosed in the mesh cage. The trajectory of mouse ambulation was video-recorded for automated post hoc analysis with the SMART video tracking system (Harvard Apparatus, Holliston, MA, USA). The time that the mice spent in the social interaction zone (30 cm × 15 cm rectangle including the mesh cage) or the social avoidance zone (30 cm × 9 cm rectangle opposite to the mesh cage) was measured. Susceptible mice were defined to spend less than 30% of their time in the social interaction zone and more than 70% in the social avoidance zone. Resilient mice were defined to spend more than 70% of their time in the social interaction zone and less than 30% in the social avoidance zone.

### 2.3. Immunohistochemistry

We conducted immunohistochemistry as described previously [11,18]. Mice were injected intraperitoneally with pentobarbital (200 mg/kg) for anesthesia. Mice were transcardially perfused with ice-cold 10% formalin (Wako Pure Chemical, Osaka, Japan). Brains were removed and incubated with 10% formalin for 24 h. Then, brains were immersed in D-PBS containing 30% sucrose and frozen in an embedding compound (FSC22, Leica, Wetzlar, Germany) on dry ice. The frozen brains were cut with a cryostat (CM 1860; Leica) at 12 or 30 μm thickness. The sections were incubated with primary antibodies for two nights at 4 °C. The primary antibodies used in this study were rat monoclonal anti-HMGB1 [8] (1:100 dilution), mouse monoclonal HMGB1 (MAB1690, R & D Systems, 1:100 dilution), mouse monoclonal αCaMKII (13-7300, Thermo Fisher Scientific, Waltham, MA, USA, 1:500 dilution), rabbit polyclonal GABA (A2052, Sigma-Aldrich, Saint Louis, MO, USA, 1:3000 dilution), rabbit polyclonal 5-HT (20080, Immunostar, 1:2000 dilution), and mouse monoclonal tyrosine hydroxylase (MAB318, Millipore, Darmstadt, Germany, 1:5000 dilution). For immunofluorescent staining, after washing with phosphate-buffered saline (PBS) containing 0.3% Triton X-100, the sections were incubated with Alexa Fluor 488-, Alexa Fluor 555-, or Alexa Fluor 647-labeled secondary antibodies (Thermo Fisher Scientific, 1:1000 dilution) for 2 h at room temperature (RT). Hoechst 33,342 (Thermo Fisher Scientific, 1:5000 dilution) was used for nuclear counterstaining. The sections were stained with NeuroTrace 530/615 for fluorescent Nissl staining. The sections were embedded in ProLong Gold Antifade Reagent (Thermo Fisher Scientific) on APS-coated glass slides (Matsunami Glass, Osaka, Japan). We acquired immunofluorescent images through a 10× objective lens (N.A. 0.45) with an epifluorescent microscope and through a 10× objective lens (N.A. 0.45) or a 40× oil-immersion objective lens (N.A. 1.25) with a confocal microscope (LSM700, Carl Zeiss Microscopy, Cambridge, UK). We analyzed HMGB1 nuclear export and the nuclear area with MetaMorph (Molecular Devices, San Jose, CA, USA). We applied predetermined thresholds for the intensity and the area to the fluorescent images to identify HMGB1-positive nuclei. The same thresholds were used in the same comparison groups. The number of HMGB1-positive nuclei per image averaged among four images taken from an indicated brain region of each mouse, normalized to the averaged value of naïve mice in each batch of experiments, was analyzed.

### 2.4. Western Blotting

Mice were anesthetized with isoflurane and transcardially perfused with ice-cold D-PBS. Brains were removed, and frontal cortices were collected as described previously [19]. The collected samples were immediately frozen in liquid nitrogen. The antibodies used in this study were rat monoclonal anti-HMGB1 [8] (1:1000 dilution), rabbit monoclonal HRP-conjugated β-actin (5125, Cell Signaling Technology, Danvers, MA, USA, 1:2000 dilution), and goat polyclonal HRP-conjugated anti-rat IgG (NA93S, Cytiva, 1:2000 dilution). The frozen samples were lysed with RIPA buffer (Fujifilm Wako) containing protease inhibitors (Sigma-Aldrich) and sonicated. After incubation on ice for 30 min, the lysed samples were centrifuged at 15,000 rpm at 4 °C for 20 min. The supernatants were collected and used in a BCA protein assay (Fujifilm Wako, Richmond, VA, USA) to determine protein concentration. Here, 10 μg of protein per lane was separated by 10% SDS-PAGE. The separated protein was transferred to PVDF membrane (Bio-Rad, Berkeley, CA, USA). The PVDF membrane was incubated with the corresponding antibodies followed by Clarity Western ECL Substrate (Bio-Rad). The bands were detected with ImageQuant 800 (Cytiva) and detected bands were quantified with ImageQuant TL software (latest version 10.2). The same samples were also analyzed with the Protein Simple assay (Bio-Techne, Minneapolis, MN, USA), a capillary electrophoresis immunoassay, according to the manufacturer’s instructions.

### 2.5. Infusion of HMGB1 Antibody and Protein into the Brain

Monoclonal HMGB1 antibody and isotype control (IgG2a) were prepared with conventional methods [8]. Recombinant HMGB1 protein was generated in Sf9 cells and purified as described previously [20]. Infusion into the brain was performed as described previously [21]. Briefly, 9-week-old male C57BL/6N mice were anesthetized with isoflurane, and the body temperature was maintained at 37 °C. For intracerebroventricular infusion, a cannula with an osmotic pump connector (Brain Infusion Kit 3, Alzet, Cupertino, CA, USA) was implanted into the right lateral ventricle (0.2 mm posterior from the bregma, 1.0 mm lateral from the midline, and 2.3 mm below the skull surface at the bregma, according to a mouse brain atlas [22], cemented in place, and then connected subcutaneously to the micro-osmotic pump (1004, Alzet). The pump delivered 22.5 ng control IgG or rat monoclonal anti-HMGB1 antibody dissolved in PBS or 5 ng recombinant HMGB1 protein or saline per day into the lateral ventricle at the flow rate of 0.11 μL/h. For local infusion into the mPFC, custom-made cannulas with an osmotic pump connector (Bio Research Center) were implanted at the mPFC bilaterally. We set the stereotaxic coordinates to the infralimbic cortex, 1.9 mm anterior from the bregma, 0.4 mm lateral from the midline, and 3.0 mm ventral from the skull surface at the bregma. After recovering from anesthesia, each mouse was transferred into a new cage and kept isolated until the behavioral experiments.

### 2.6. Statistical Analyses

Data are shown as mean ± SEM. We analyzed the comparison of two groups with a two-sided unpaired *t*-test. Comparison of three or more groups was analyzed by either two-way ANOVA (Figure 1b, Figure 2b, Figure 3b, Figure 4b, Figure 5f, and Figure 6b) with one factor as a repeated measure or one-way ANOVA (Figure 6d and Appendix A), followed by Tukey’s multiple comparison test. We used PRISM 9 software (GraphPad, San Diego, CA, USA) for these analyses. *p* values less than 0.05 were considered significant.

## 3. Results

### 3.1. Extracellular HMGB1 from Prefrontal Neurons Promotes Chronic Stress-Induced Social Avoidance

It was reported that acute intracerebroventricular (i.c.v.) HMGB1 infusion induces depression-like behaviors under acute stress, such as prolonged immobility with tail suspension and decreased sucrose preference [14]. Using a mouse model of repeated social defeat stress, we examined whether i.c.v. HMGB1 infusion would promote depression-related behaviors induced by chronic stress. We continuously infused recombinant human HMGB1 protein into the lateral ventricles from 1 week prior to repeated social defeat stress (Figure 1a). Unexpectedly, this HMGB1 infusion attenuated social avoidance, one of the depression-related behaviors induced by repeated social defeat stress, compared with the vehicle infusion (Figure 1b). Given that repeated social defeat stress may cause an extracellular release of endogenous HMGB1 proteins, we also examined whether the blockade of extracellular HMGB1 would affect chronic stress-induced depression-related behaviors. We continuously infused HMGB1-neutralizing antibodies or control antibodies into the lateral ventricles from 1 week prior to repeated social defeat stress (Figure 2a). The HMGB1 antibodies infused into the lateral ventricles did not inhibit, but rather promoted, social avoidance induced by repeated social defeat stress, compared with the control antibodies (Figure 2b). Thus, contrary to previous reports suggesting its pro-depressive role, our experiments suggested that endogenous extracellular HMGB1 in the brain could suppress chronic stress-induced depression-like behaviors.

**Figure 1 cells-12-01789-f001:**
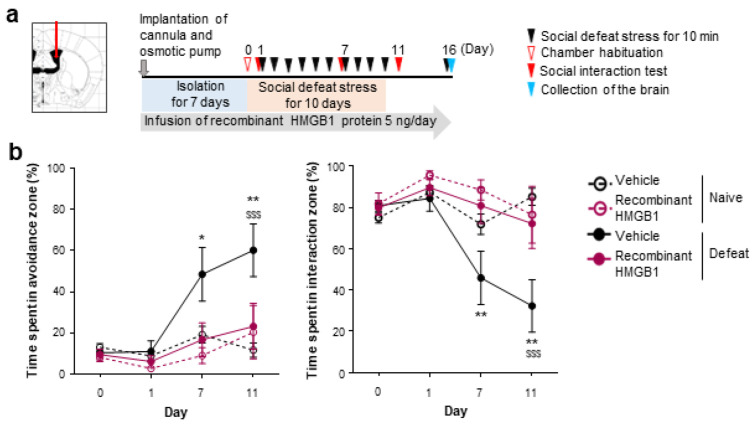
Intracerebroventricular infusion of recombinant HMGB1 suppresses repeated social defeat stress-induced social avoidance. (**a**) A scheme of the behavioral experiment. (**b**) The inhibitory effect of recombinant HMGB1 protein on the development of social avoidance by repeated social defeat stress. Mice in the Defeat group, which received the stress, and the Naïve group, which did not, were implanted with a cannula at the right lateral ventricle connected to an osmotic pump placed in the subcutaneous tissue of the back. The mice were kept isolated for 7 days before the stress (Day 1 to Day 10). Recombinant HMGB1 protein (5 ng/day) or saline (vehicle) had been infused throughout the experimental period from the cannula implantation. Data are shown as mean ± SEM. * *p* < 0.05, ** *p* < 0.01 between the mice infused with vehicle and recombinant HMGB1 protein in the Defeat group. $$$ *p* < 0.001 between the Naïve and Defeat groups with vehicle infusion.

**Figure 2 cells-12-01789-f002:**
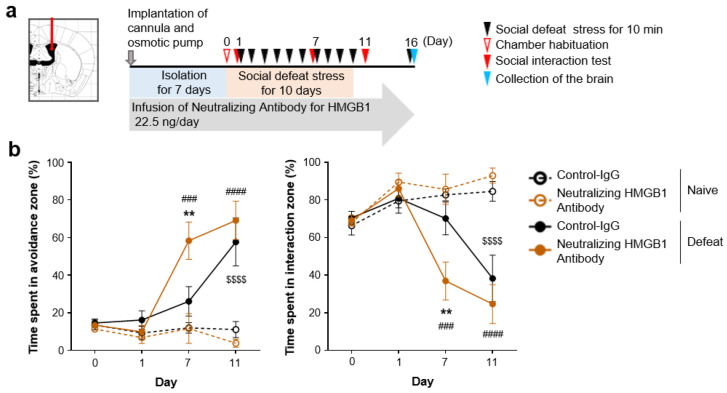
Intracerebroventricular infusion of neutralizing HMGB1 antibody promotes repeated social defeat stress-induced social avoidance. (**a**) A scheme of the behavioral experiment. (**b**) The facilitatory effect of neutralizing HMGB1 antibody on the development of social avoidance by repeated social defeat stress. Mice in the Defeat and Naïve groups were implanted with the cannula and osmotic pump, and neutralizing HMGB1 antibody (22.5 ng/day) or isotype control IgG was infused to the right lateral ventricles, as described in the legend of Figure 1. Data are shown as mean ± SEM. ** *p* < 0.01 between the mice infused with control IgG and neutralizing HMGB1 antibody in the Defeat group. $$$$ *p* < 0.0001 between the Naïve and Defeat groups with control IgG infusion. ### *p* < 0.001, #### *p* < 0.0001 between the Naïve and Defeat groups with neutralizing HMGB1 antibody infusion.

However, as TLR2/4-mediated microglial activation in the mPFC is crucial for social avoidance induced by repeated social defeat stress, extracellular HMGB1 in selective brain regions, such as the mPFC, could promote depression-like behaviors. To test this possibility, we examined whether the blockade of extracellular HMGB1 in the mPFC would ameliorate social avoidance induced by repeated social defeat stress. We continuously infused HMGB1-neutralizing antibodies (at the same concentration as those infused to the lateral ventricle described above) or control antibodies to the mPFC from 1 week prior to repeated social defeat stress (Figure 3a). Repeated social defeat stress induced social avoidance in the presence of the control antibodies, but this behavior was not significantly induced in the presence of the HMGB1 antibodies (Figure 3b). As optogenetic stimulation of cultured neurons reportedly releases HMGB1 to the extracellular space [23], we examined whether neuronal HMGB1 was involved in the pro-depressive action of HMGB1 in the mPFC. To delete HMGB1 in αCaMKII-positive forebrain neurons, including mPFC excitatory neurons, we obtained male littermates obtained from crossing HMGB1fl/fl (control) and αCaMKII-CreERT/+; HMGB1fl/fl (forebrain neuron-specific HMGB1 knockout) mice, followed by tamoxifen injection. Three weeks after the tamoxifen injection, these mice were subjected to the behavioral experiment (Figure 4a). Repeated social defeat stress induced depression-related behavior in the control littermates, but this behavior was not significantly induced in forebrain neuron-specific HMGB1 knockout mice (Figure 4b). After the behavioral experiment, we confirmed HMGB1 deletion in αCaMKII-positive forebrain neurons, including mPFC excitatory neurons, of the HMGB1 knockout mice (Figure 4c,d).

**Figure 3 cells-12-01789-f003:**
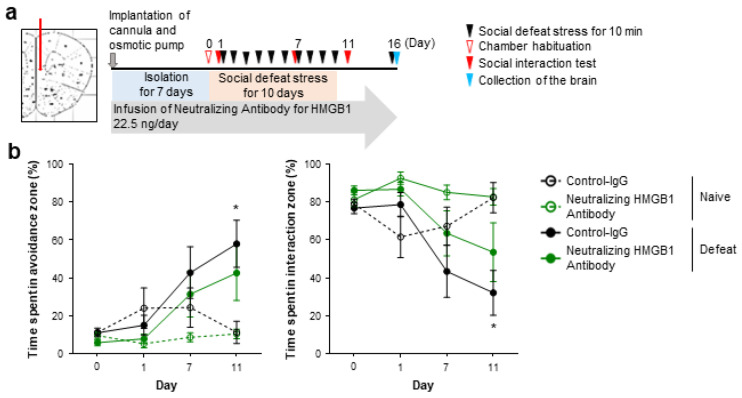
mPFC infusion of the neutralizing HMGB1 antibody attenuates repeated social defeat stress-induced social avoidance. (**a**) A scheme of the behavioral experiment. (**b**) The inhibitory effect of neutralizing HMGB1 antibody on the development of social avoidance by repeated social defeat stress. Mice in the Naïve and Defeat groups were implanted with cannulas bilaterally at the mPFC connected to an osmotic pump placed in the subcutaneous tissue of the back. The mice were kept isolated for 7 days before the stress (Day 1 to Day 10). Neutralizing HMGB1 antibody (22.5 ng/day) or control IgG had been infused throughout the experimental period from the cannula implantation. Data are shown as mean ± SEM. * *p* < 0.05 between the Naïve and Defeat groups with control IgG infusion.

Collectively, these findings suggest that extracellular HMGB1 released from mPFC excitatory neurons promotes chronic stress-induced depression-related behaviors.

**Figure 4 cells-12-01789-f004:**
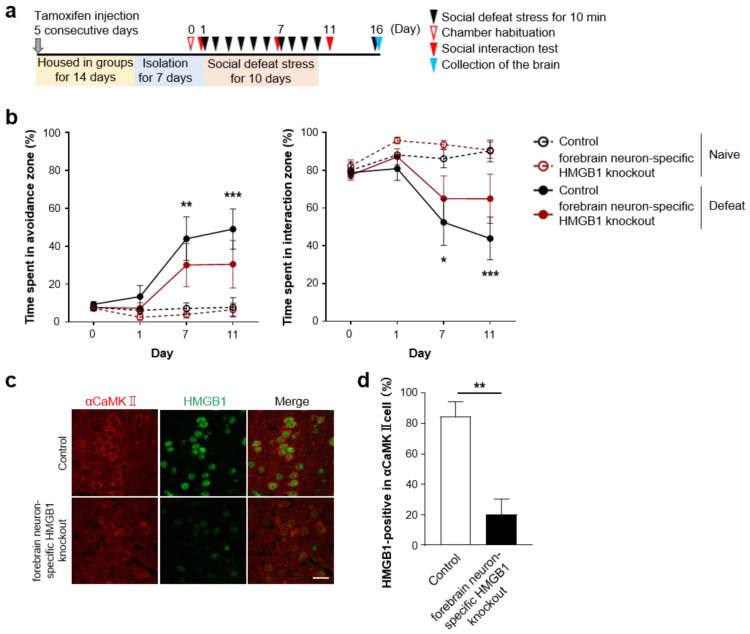
HMGB1 deletion in αCaMKII-expressing neurons attenuates repeated social defeat stress-induced social avoidance. (**a**) A scheme of the behavioral experiment. (**b**) The inhibitory effect of HMGB1 deletion in αCaMKII-expressing neurons on the development of social avoidance by repeated social defeat stress. Control and forebrain neuron-specific HMGB1 knockout mice (HMGB1fl/fl and αCaMKII-CreERT/+; HMGB1fl/fl, respectively) were obtained by tamoxifen injection for 5 consecutive days. The mice were group-housed for 14 days and isolated for 7 days before the stress (Day 1 to Day 10). For histological analyses in (**c**,**d**), the mice were sacrificed at 90 min after an additional social defeat stress on Day 16. Data are shown as mean ± SEM. * *p* < 0.05, ** *p* < 0.01, *** *p* < 0.001 between the Naïve and Defeat groups of control (HMGB1fl/fl) mice. (**c**,**d**) HMGB1 deletion in αCaMKII-expressing neurons. Representative images (**c**) show HMGB1 immunofluorescent signals in αCaMKII-positive neurons in the mPFC in control and forebrain neuron-specific HMGB1 knockout mice (scale bar, 20 μm). The percentage of HMGB1-positive nuclei in αCaMKII-positive neurons is shown (**d**). Data are shown as mean ± SEM. ** *p* < 0.01 for unpaired *t*-test.

### 3.2. Chronic Stress Induces HMGB1 Nuclear Export in Prefrontal Neurons in a RAGE-Dependent Manner

As our finding suggested that HMGB1 released from mPFC excitatory neurons is involved, we performed immunofluorescent staining to examine whether repeated social defeat stress would change HMGB1 localization in the mPFC (Figure 5a). Two independently generated antibodies to HMGB1 which recognized different epitopes showed colocalized signals in the mPFC (Appendix A). Before the stress, most αCaMKII-positive excitatory neurons expressed HMGB1, whereas only half of GABAergic inhibitory neurons did so (Figure 5e,f). Repeated social defeat stress decreased the number of HMGB1-positive nuclei in the mPFC immediately (Appendix A) and at 90 min (Figure 5b–d) after the last stress exposure. We performed Western blotting of HMGB1 in mPFC whole-tissue lysates but did not find a difference in its signal intensity between the Naïve and Defeat groups (Appendix A), suggesting that HMGB1 proteins are retained in the mPFC after repeated social defeat stress despite its nuclear export and putative extracellular release. Nuclear HMGB1 signals were recovered in 24 h (Appendix A). The HMGB1 changes were heterogeneous and, thus, prominent in a small subregion, within the mPFC (Figure 5b), although their distribution and extent varied among individual mice. As there is large individual variability of stress susceptibility, stressed mice could be categorized to susceptible mice and resilient mice, depending on the level of social avoidance. However, the HMGB1 changes occurred in both groups of mice (Figure 5d). The decrease in nuclear HMGB1 signals occurred in excitatory and inhibitory neurons (Figure 5e,f). Further histological analyses revealed the nuclear shrinkage of mPFC neurons induced by repeated social defeat stress (Appendix A), suggesting neuronal damages. 

**Figure 5 cells-12-01789-f005:**
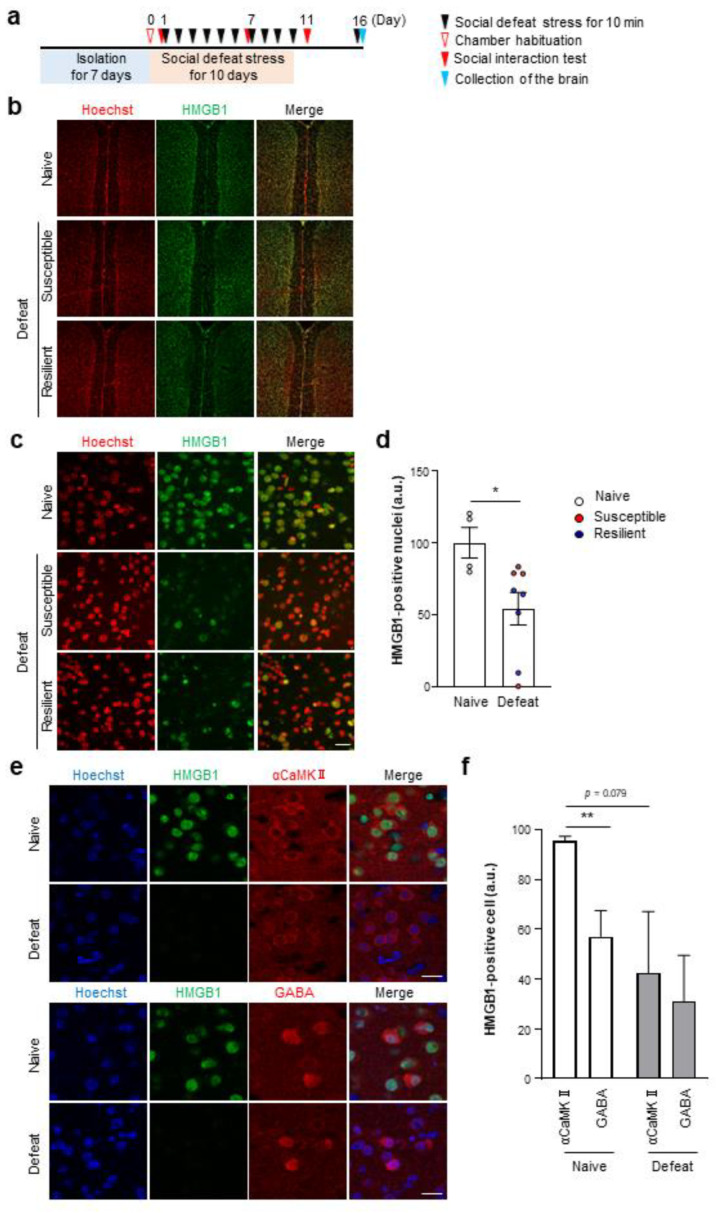
Repeated social defeat stress induces HMGB1 nuclear export in the mPFC of both susceptible and resilient mice. (**a**) A scheme of the behavioral experiment. The Naïve and Defeat mice were kept isolated for 7 days before repeated social defeat stress (Day 1 to Day 10). The mice were sacrificed at 90 min after an additional social defeat stress on Day 16 for HMGB1 immunostaining. Susceptible and resilient mice were chosen from the Defeat mice according to the behavioral criteria described in the Methods section. (**b**,**c**) Representative images of HMGB1 immunostaining in the mPFC of Naïve, susceptible, and resilient mice at low and high magnification, respectively. HMGB1 signals are shown in green. Nuclei were counterstained with Hoechst (red). Scale bar, 100 μm in (**b**) and 20 μm in (**c**). (**d**) The number of HMGB1-positive nuclei in the mPFC in the Naïve and Defeat groups. Individual dots represent the value of each mouse normalized to the average of the Naïve mice in each batch of experiments. Red and blue circles show the values of susceptible and resilient mice, respectively. Data are shown as mean ± SEM. * *p* < 0.05 for unpaired *t*-test. (**e**) Representative images of HMGB1 co-immunostaining with αCaMKII and GABA in the mPFC of Naïve and Defeat mice. HMGB1 signals are shown in green. αCaMKII and GABA are shown in red. Nuclei were counterstained with Hoechst (blue). Scale bar, 20 μm. (**f**) The percentage of HMGB1-positive nuclei in αCaMKII- or GABA-positive mPFC neurons of Naïve and Defeat mice. Data are shown as mean ± SEM. ** *p* < 0.01 between αCaMKII and GABA in the Naïve group.

Notably, HMGB1 knockout in αCaMKII-positive forebrain neurons abolished the nuclear shrinkage of neurons (Appendix A), suggesting that neuronal HMGB1 contributes to the neuronal damages. By contrast, repeated social defeat stress did not alter nuclear HMGB1 signals in the other brain regions we examined, such as the nucleus accumbens, ventral tegmental, amygdala, hippocampus, dorsal raphe nucleus, and locus coeruleus, at least in 90 min (Appendix A). These findings show that repeated social defeat stress transiently induces HMGB1 nuclear export selectively in mPFC neurons associated with local histological damages.

Since HMGB1 binds to RAGE to augment innate immune signaling [24], we examined the involvement of RAGE in repeated social defeat stress using RAGE-KO mice (Figure 6a). RAGE deficiency abolished HMGB1 nuclear export in mPFC neurons after repeated social defeat stress (Figure 6c,d), suggesting the positive feedback loop of HMGB1-RAGE signaling. However, RAGE deficiency did not affect repeated social defeat stress-induced social avoidance (Figure 6b). Thus, the HMGB1 nuclear export in mPFC neurons was dispensable for the development of social avoidance, although extracellular HMGB1 in the mPFC promoted social avoidance.

**Figure 6 cells-12-01789-f006:**
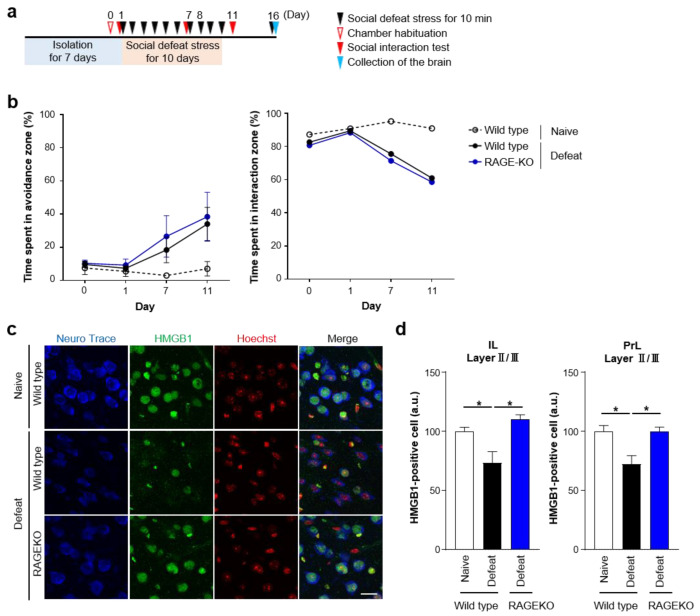
RAGE deletion inhibits repeated social defeat stress-induced HMGB1 nuclear export. (**a**) A scheme of the behavioral experiment. Wild-type and RAGE-KO mice in the Naïve and Defeat group were subjected to behavioral experiments as described in the legend of Figure 5. (**b**) No effect of RAGE deletion on the development of social avoidance by repeated social defeat stress. (**c**,**d**) The inhibitory effect of RAGE deletion on HMGB1 nuclear export induced by repeated social defeat stress. Representative images (**c**) show HMGB1 immunofluorescent signals in the mPFC in wild-type and RAGE-KO mice of the Defeat group as well as wild-type mice of the Naïve group. HMGB1 signals are shown in green. Brain slices were counterstained with Neurotrace and Hoechst shown in blue and red, respectively. Scale bar, 20 μm. The number of HMGB1-positive nuclei of neurons in layers II/III of the mPFC (infralimbic (IL) and prelimbic (PrL) subregions, separately), normalized to the average of the Naïve mice in each batch of experiments, is shown (**d**). * *p* < 0.05 for Tukey’s multiple comparison for the indicated pairs.

## 4. Discussion

With the advent of neutralizing antibodies and conditional knockout mice, we examined the behavioral roles of endogenous extracellular HMGB1 in the brain under chronic stress. The infusion of HMGB1-neutralizing antibodies to the mPFC attenuated repeated social defeat stress-induced social avoidance, one of depression-related behaviors. Furthermore, genetic deletion of HMGB1 in αCaMKII-positive forebrain neurons including mPFC excitatory neurons similarly attenuated it. These findings show, for the first time, that endogenous extracellular HMGB1 derived from mPFC excitatory neurons contributes to chronic stress-induced depression-related behavior.

Recent studies have found that the redox state of HMGB1 is a key determinant of its receptor interaction and immunological function [25]. HMGB1 has three cysteines at its 23rd, 45th, and 106th amino acid residues. The fully reduced form of HMGB1 is thought to form a heterodimer with CXCL12 and act on CXCR4 to promote mononuclear cell recruitment to the injured tissue [26]. The fully reduced HMGB1 is also known to bind to RAGE. By contrast, the disulfide form of HMGB1 with a disulfide bridge between Cys23 and Cys45 and reduced Cys106 induces transcription and secretion of proinflammatory cytokines in macrophages through binding to TLR4 [27]. The oxidized form appears to lose these biological activities. Since TLR2/4 in mPFC microglia are crucial for repeated social defeat stress-induced social avoidance [18], extracellular HMGB1 derived from mPFC neurons could take the disulfide form and exert pro-depression-like actions through TLR2/4-mediated microglial activation. It was reported that a persistent state of oxidative stress results in stress-induced susceptibility to depression [28]. Furthermore, in rodents, chronic corticosterone treatment induces oxidative stress in the mPFC along with depression-like behaviors via NADPH oxidase 1 [29]. Stress-induced oxidative stress could convert the reduced form of HMGB1 to its disulfide form in the mPFC. Indeed, the oxidative stress could underlie the HMGB1-dependent nuclear shrinkage of mPFC neurons caused by repeated social defeat stress, as oxidative stress is reportedly accompanied by nuclear shrinkage with oxidative DNA damage [30]. Since antibody blockade of HMGB1 in the mPFC and genetic deletion of HMGB1 in αCaMKII-positive forebrain neurons only partially attenuated the depression-like behavior, other TLR2/4 ligands could exist in the mPFC to compensate for the lack of the pro-depressive HMGB1 actions. We previously reported that repeated social defeat stress increases mRNA expression of S100A8 and S100A9, which form a heterodimer as one of the putative TLR2/4 ligands, in the mPFC [18]. Whether other TLR2/4 ligands, such as S100A8/A9, exert pro-depression-like actions together with HMGB1 in the mPFC remains to be examined.

We found that repeated social defeat stress induces HMGB1 nuclear export selectively in mPFC neurons. Since HMGB1 nuclear export occurred immediately after stress exposure, it should move to the cytoplasm first. However, cytoplasmic HMGB1 signals did not appear to increase in our experiments. This lack of effect on cytoplasmic HMGB1 signals may be due to its subsequent extracellular release. Biochemical methods may be useful to address this issue, but are difficult to apply to this case, as the regions showing HMGB1 nuclear export within the mPFC vary considerably among individual stressed mice. The nuclear export of HMGB1 by activated macrophages and dendritic cells is controlled by its posttranslational modifications, such as acetylation in several lysine residues near one of its nuclear localization signals [31]. Such modifications could underlie stress-induced HMGB1 nuclear export in mPFC neurons. As optogenetic stimulation reportedly induces extracellular HMGB1 release from cultured neurons [23], stress-induced excitation of mPFC neurons could cause HMGB1 nuclear export. Repeated social defeat stress activates neurons in many brain areas, not only in the mPFC, but also in the nucleus accumbens, hippocampus, amygdala, and locus coeruleus, where HMGB1 nuclear export was absent. Thus, stress-induced neuronal activity alone cannot account for the mPFC selectivity of the HMGB1 nuclear export. Unexpectedly, RAGE, which can be activated by HMGB1, is required for the HMGB1 nuclear export of mPFC neurons after repeated social defeat stress. Thus, HMGB1 released to the extracellular space could act on RAGE, thereby amplifying HMGB1 release by a positive feedback loop. As multiple endogenous ligands for RAGE exist besides HMGB1 [32], another RAGE ligand(s) could also contribute to stress-induced HMGB1 nuclear export in mPFC neurons. Given that antibody blockade of HMGB1 suggested the role of extracellular HMGB1 for repeated social defeat stress-induced social avoidance, HMGB1 should further be released to the extracellular space. Gasdermins, pore-forming effector proteins that cause membrane permeabilization, have been suggested to allow HMGB1 to penetrate the plasma membrane for their extracellular release [33]. As gasdermin D is reportedly crucial for chronic mild stress-induced depression-like behavior [34], it could mediate extracellular HMGB1 release from mPFC neurons upon repeated social defeat stress. 

Unexpectedly, using infusions of HMGB1-neutralizing antibodies and recombinant protein into the cerebral ventricles, we discovered antidepressive-like actions of extracellular HMGB1 in the brain, in addition to the pro-depressive-like actions suggested previously. These antidepressive-like actions could be mediated by yet-unidentified brain regions via HMGB1 receptors other than TLR2/4. Given that RAGE deletion abolished HMGB1 nuclear export in mPFC neurons but spares the concomitant social avoidance, RAGE could be involved in both pro-depressive and antidepressive actions of HMGB1. Alternatively, CXCR4 is reportedly involved in enriched environment-induced hippocampal neurogenesis that mediates the antidepressive-like actions at least in part [35]. Whether HMGB1 exerts antidepressive-like actions through RAGE and/or CXCR4 is an interesting possibility to be investigated.

In conclusion, we showed the roles of endogenous extracellular HMGB1 from mPFC neurons in regulating chronic stress-induced depression-related behavior. However, since our findings also suggest multiple, potentially counteracting, roles of HMGB1 in this behavioral change, a further understanding of HMGB1 regulations and actions is warranted to exploit these findings for drug development for depression. Given a higher proportion of stress-related mental illness, such as depression, in females over males, whether the behavioral role of HMGB1 in mPFC neurons found in this study is also applied to female mice warrants future investigation.

## Data Availability

Not applicable.

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
