# Peer review of "Repeated Social Defeat Stress Induces HMGB1 Nuclear Export in Prefrontal Neurons, Leading to Social Avoidance in Mice"

_cells, 2023, doi:10.3390/cells12131789_

Round 1

Reviewer 1 Report (Previous Reviewer 1)

Dear authors; 

Thank you for your corrections. 

This article is valuable The article can be published in the journal after the revisions are made.

Reviewer 2 Report (Previous Reviewer 2)

I appreciate for the efforts of authors in revision, and the additional data have addressed my concern. I recommend acceptance of the revised manuscript.

This manuscript is a resubmission of an earlier submission. The following is a list of the peer review reports and author responses from that submission.

Round 1

Reviewer 1 Report

1.  The main question addressed by the research: What behavioral function does endogenous extracellular HMGB1 perform when chronic stress is present?

2. Previous research demonstrated the functions of endogenous HMGB1 in depression-related behaviors because it only demonstrated the pro-depressive effects of exogenously applied HMGB1 and the anti-depressive effects of HMGB1 inhibitors that bind to and inhibit HMGB1 as well as other metabolic enzymes.

This study is significant because it examines the function of endogenous extracellular HMGB1 from mPFC neurons in regulating behaviors linked to repeated social defeat stress, such as depression caused by chronic stress.

3. Previous research concentrated on the involvement of HMGB1 (neutralizing antibodies or antagonists) in brain injury in animal models. Acute stress-related behaviors were assessed in all prior research investigations for depression. However, this study is a thorough investigation that discussed the influence of endogenous extracellular HMGB1, which is produced by excitatory neurons in the mPFC, on chronic stress-related behaviors like depression.

4. Methodology: A combination of immunohistochemistry, repeated social defeat stress, a social interaction test, and the infusion of HMGB1 antibody and protein into the brain was used to determine how repeatedly experiencing social defeat stress transiently increases HMGB1 nuclear export selectively in mPFC neurons associated with local histological abnormalities. Connexin36 and kynurenine pathway gene expression measurement is advised as a next step in the inquiry.

5. The conclusion section of the paper is comprehensive, accurate, and completely expressive.

6. The citations were correct and pertinent to the subject.

7. The tables and figures are presented properly and, in my opinion, do not require any additional adjustments.

Author Response

We thank this reviewer for highly evaluating our manuscript. We also appreciate your thoughtful suggestion on our next step to test the possible roles of connexin36 and kynurenine pathway. We look forward to analyzing it in our future study.

Reviewer 2 Report

The authorse explored the roles of endogenous HMGB1 under chronic stress in mice, and provided some new information. The study is well design and written. Some points need to be revised.

1. Why choose male mice only in the study?

2. Could the author add quantative menthods such as WB to detect the protein expression of HMGB1, αCaMKII in figure 4, 5, 6?

3. the seperate band of each group in figure 5b, 5e need to be revised.

4. In the method, author mentioned have used rabbit polyclonal 5-HT, mouse monoclonal tyrosine hydroxylase, I have not seen the results.

Author Response

We greatly thank this reviewer for highly evaluating our manuscript.

  1. Why choose male mice only in the study?
    Response: We thank this reviewer for this insightful comment. The repeated social defeat stress model relies on the natural tendency of male resident mice to attack male intruder mice. Although several attempts have been reported to apply this model to female mice, a reliable protocol has not been established. Nonetheless, given a higher proportion of depression in females than males, we should investigate whether the findings in male mice in this study can be reproduced in female mice in future studies. We added this consideration in the Discussion section below (Lines 406-409).
    “Given a higher proportion of stress-related mental illness, such as depression, in females than males, whether the behavioral role of HMGB1 in mPFC neurons found in this study is also applied to female mice warrants future investigation.”
  2. Could the author add quantitative methods such as WB to detect the protein expression of HMGB1, αCaMKII in figure 4, 5, 6?
    Response: We agree with the importance of the suggested biochemical analyses to track the movement of HMGB1 proteins to the cytoplasm and extracellular space after the nuclear export induced by stress. However, the size and location of the brain regions having HMGB1 nuclear export in the medial prefrontal cortex vary considerably among individual stressed mice. Thus, it is difficult to precisely excise the target brain regions to analyze for quantitative biochemical analyses. We added this consideration in the Discussion section below (Lines 363-368).
    “Since HMGB1 nuclear export occurred immediately after stress exposure, it should move to the cytoplasm first. However, cytoplasmic HMGB1 signals did not appear to increase in our experiments. This lack of effect on cytoplasmic HMGB1 signals may be due to its subsequent extracellular release. Biochemical methods may be useful to address this issue, but are difficult to apply to this case, as the regions showing HMGB1 nuclear export within the mPFC considerably vary among individual stressed mice.
  3. the separate band of each group in figure 5b, 5e need to be revised.
    Response: As suggested, we put gaps between the images in the revised Figure 5b and 5e.
  4. In the method, author mentioned have used rabbit polyclonal 5-HT, mouse monoclonal tyrosine hydroxylase, I have not seen the results.
    Response: Since these antibodies were used for the immunohistochemistry shown in Supplementary Figure 4a, we have kept the descriptions of the antibodies.

Round 2

Reviewer 2 Report

The authors have not addressed my concern in the revised version.